# Rényi Entropy-Based Adaptive Integration Method for 5G-Based Passive Radar Drone Detection

Radosław Maksymiuk *, Karol Abratkiewicz, Piotr Samczyński and Marek Płotka

Institute of Electronic Systems, Faculty of Electronics and Information Technology, Warsaw University of Technology, 00-665 Warsaw, Poland
* Correspondence: radoslaw.maksymiuk@pw.edu.pl

**Abstract:** This paper presents the first successful drone detection results using a 5G network as a source of illumination in a passive radar system. Furthermore, a novel adaptive strategy for signal integration is shown. The proposed approach is based on the Rényi entropy. It allows one to select time frames with a densely allocated downlink channel both in the time and frequency domains. The resource allocation is strongly related to a network load and has a crucial influence on 5G-based passive radar range resolution and detection capabilities. The proposed technique was validated using simulated and real-life signals, confirming the possibility of detecting unmanned aerial vehicles (UAVs) in 5G-network-based passive radars. Moreover, the proposed methodology can be directly used in passive radar systems where the illuminating signal duration and bandwidth are content-dependent, and the radar resolution may vary significantly.

**Keywords:** passive bistatic radar (PBR); 5G; passive coherent location (PCL); entropy; UAV

## 1. Introduction

Current research on different sources of illumination for passive radars has been very intense. Besides well-known illuminators such as digital video broadcasting—terrestrial (DVB-T) [1,2], digital video broadcasting—satellite (DVB-S) [3], satellites in geosynchronous orbits [4], the Global Navigation Satellite System (GNSS) [5], frequency-modulated (FM) analogue radio [6,7], or digital audio broadcasting (DAB) [2,8] radios, one may note several novel ideas involving 5G [9], long-term evolution (LTE) [10] and Starlink [11–13] transmission for target detection. Theoretical studies and simulations show that these sources of illumination are suitable for passive radar purposes. Recent experiments have proved the ability to detect large targets, such as cars, using newly deployed telecommunications standards, e.g., the 5G network [9,14]. However, the continuous dynamic growth of the number of unmanned aerial vehicles (UAVs) has created a demand for drone detection capability (flight control and security). Passive radar based on a 5G network is a promising technology regarding the satisfaction of this demand. It is characterized by low cost and easy deployment thanks to there being no need for it to have its own transmitter and transmission license. 5G networks provide excellent coverage of base stations and a relatively wide band, therefore, one can detect small targets such as UAVs. The latter parameter (signal bandwidth) is of particular interest, since it allows for the obtaining of high-range resolution. For example, in Poland, the 5G network will cover up to 80 MHz from a single telecommunications provider, which gives up to 400 MHz for five expected providers [9]. This excellent value gives a maximum range resolution of 75 cm and, in the future, it certainly could be used for radar imaging and micro-Doppler signature analysis.

Some experiments covering the utilization of a 5G network for passive radar UAV detection have already been conducted. In [15], the authors describe simulations of LTE and 5G New Radio (5G NR)-based radar and experiments with successful drone (holding a multi-faceted reflector) detection using synthetically generated waveforms. In [16], a concept of using 5G base stations as a millimeter wave active radar system capable of

detecting and identifying UAVs was presented. However, the authors did not consider the specific 5G waveform, and the simulation was carried out without assumptions on both signal and network properties (content-dependent transmission). Moreover, the simulations in [16] comprised only the parameter estimation of the signal reflected from a drone (micro-Doppler analysis), neglecting the problem of target detection in a practical scenario. Successful UAV detection utilizing a software-defined radio (SDR) transmitter was also reported in [17]. In [9], a passive detection of a car was obtained employing the same cooperative 5G network as in the present article. In [9], many challenges and difficulties connected with 5G passive radar systems were addressed, but none of them were solved. One problem in 5G-based passive radar is a content-dependent transmission that disables target detection when the base station does not emit any downlink signal. Furthermore, there have still not been any experiments reported with a real-life and operational 5G network used as a source of illumination for passive drone detection. Thus, this topic is the main novelty presented in this paper.

Content-dependent transmission is the main problem limiting the possibility of using passive radar based on illumination from the 5G network. As a result, the illumination has a variable filling both in time and frequency. It may happen that in the absence of data to be transmitted via downlink, the amount of signal is reduced to a minimum. Furthermore, in the case of a partial resources allocation, there may be situations where the full downlink allocation will be temporary, and the rest of the time it will use only small portions of frequency. Then, the range resolution will be limited and variable, which renders inferior the properties of the passive radar. Only when the network resources are fully used is it possible, for example, to detect small objects such as drones. Thus, there is a clear need for an algorithm that allows the selection of time intervals with a dense allocation of resources in time and frequency. Thanks to this, it will be possible to make the results of radar processing independent of the momentary drops in the energy of the signal emitted by the 5G network.

Content-dependent transmission is nothing new, as it is found in other systems such as WiFi [18–20] and FM radio [21,22]. In [21], an approach for FM transmitter selection is proposed. It is based on instantaneous effective bandwidth measurement. It is evaluated in this paper in regard to using 5G networks as illuminators of opportunity for passive radar, as a reference method. In other works so far, especially those describing the possibilities of using the 5G network for target detection, no studies show how to adaptively select the time interval for signal integration. Usually, the work comes down to manual signal part selection with suitable parameters from the point of view of radiolocation. In [15–17], 5G signals with a full amount of content were employed, and the content dependency issue was not addressed. In [9], this topic was highlighted but was not solved. Thus, this paper is the first (to the best of the authors' knowledge) to address and solve this issue in a 5G-based passive radar system. Future 5G-network-based passive radars could be used to secure critical infrastructure, so adaptive algorithms should be provided to decide on the right moment of integration.

## 2. 5G Passive Radar Principles

Passive bistatic radar (PBR) or Passive Coherent Location (PCL), unlike active radar, does not have its own source of illumination. It relies on the usage of a non-cooperating transmitter of opportunity. Typically, the transmitter is in a different location to the receiver, which leads to bistatic geometry, depicted in Figure 1. The passive radar receives an electromagnetic wave directly from the transmitter ($x_r$—reference signal) and an echo reflected from the target ($x_s$—surveillance signal). Therefore, it requires at least two receiving channels working coherently. Directional antennas or antenna arrays are usually used for those channels in order to employ beamforming techniques. The signals are coherently received and digitized, yielding $x_{\mathrm{ref}}$—from the reference channel, and $x_{\mathrm{sur}}$—from the surveillance channel.

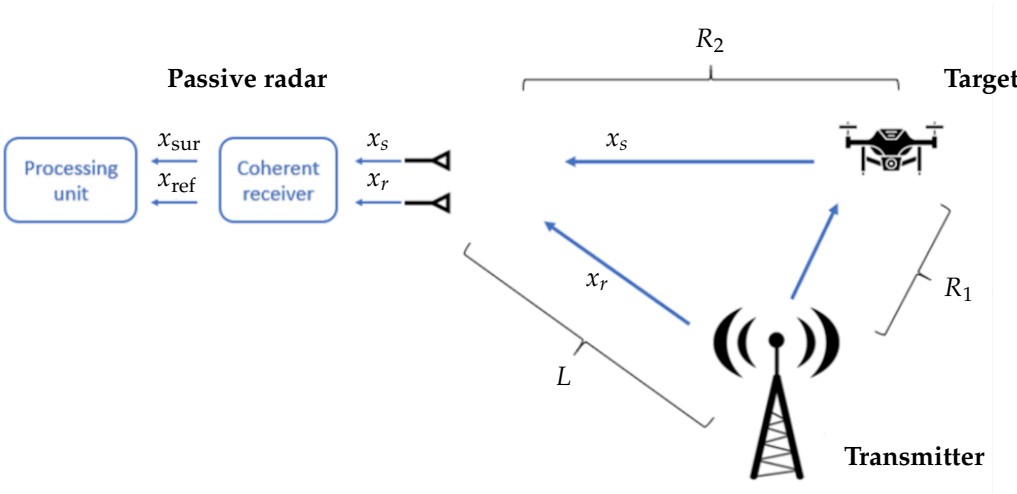

**Figure 1.** Typical PCL geometry.

The aim of passive bistatic radar is to determine the bistatic range and bistatic velocity of a target [23]. The bistatic range $R_b$ is defined as the difference of the indirect path (with reflection of the target) and direct path of the wave:

$$R_b = R_1 + R_2 - L, \tag{1}$$

where $R_1$ is the range from the transmitter to the target, $R_2$ is the range from the target to the receiver, and $L$ is the baseline length (see Figure 1). The bistatic velocity $V_b$ is obtained from the measured Doppler shift between $x_{\text{ref}}$ and $x_{\text{sur}}$ and is defined as

$$V_b = -\lambda f_d, \tag{2}$$

where $\lambda$ is the wavelength of the signal and $f_d$ is the measured Doppler shift. Once the signals are received and digitized, they are then digitally processed. A typical PCL processing scheme is depicted in Figure 2.

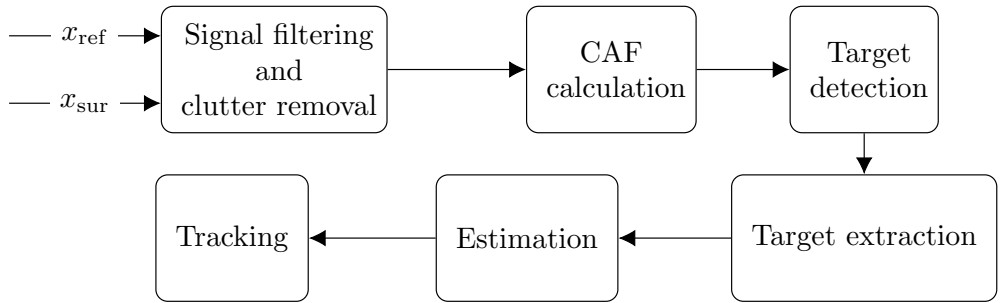

**Figure 2.** PCL radar processing scheme.

Here, the authors intentionally omitted the step that is usually the first one in the processing pipeline, namely digital beamforming (DBF) [24]. The main rationale behind DBF is to provide a highly selective reference beam (so that echoes from observed targets are not present inside). This provides a larger radar surveillance area. In addition, using DBF, the direction in which a target is located relative to the radar can be estimated. This is additional information that can be used to improve the quality of target tracking or target localization. However, since the authors used a simplified demonstrator with only two antennas during the presented experiments, the DBF technique was abandoned and not included in Figure 2.

Passive radars operate with very low energy of echoes coming from targets. To observe echoes from targets, it is necessary to remove all unwanted components present in the surveillance signal $x_{\text{sur}}$. These components result from the leakage of the transmitted signal, and are called direct-path interference (DPI); they can also come from echoes from stationary or slowly moving targets. Adaptive filters or the CLEAN method are used to remove them [25,26].

To obtain the bistatic range and velocity of the target, the cross-ambiguity function (CAF) is computed with the following formula [23]:

$$\chi(R_b, V_b) = \int_{-T_{\text{int}}/2}^{T_{\text{int}}/2} x_{\text{sur}}(t) x_{\text{ref}}^* \left( t - \frac{R_b}{c} \right) e^{-j2\pi \frac{V_b}{\lambda} t} \, \mathrm{d}t, \tag{3}$$

where $T_{\text{int}}$ is the integration time.

For the purpose of carrying out target detection, a thresholding of the CAF against its noise estimate is performed [23]. This assumes a certain minimum signal-to-noise ratio (SNR) for which false and true detections are fixed (with given values of the probability of false alarm $P_{\text{fa}}$ and probability of detection $P_d$, respectively). The simplest estimator of the noise level is a constant value estimator, such as the mean or median of the CAF values. However, the noise distribution cannot always be modeled in this way, and more sophisticated adaptive methods are needed, such as the constant false alarm rate (CFAR) processor [27].

After the thresholding operation, a set of detections is obtained. Usually, the echo from a single target is spread over several adjacent range and velocity cells, where the midpoint typically marks the correct position estimation of the target. For this reason, it is necessary to perform an extraction operation, which consists of grouping the CAF cells corresponding to a single detection. Methods known from image processing techniques, i.e., clustering, as well as morphological operations of opening or closing, can be used here [28].

When individual CAF cells are grouped into sets of detections, the parameters of each can be estimated. In the simplest case, the distance and bistatic velocity of the CAF cell with the largest amplitude value can be taken. However, in this case, the accuracy of measurement of these parameters is identical to the resolution of the bistatic distance and velocity. To improve the estimation, it is possible to use a fairly simple estimator with curve fitting, such as a parabola [29]. With a sufficiently high detection SNR (about 20 dB), it is possible to improve the accuracy of bistatic distance measurement by up to 10 times over the assumed measurement resolution [23].

The final stage of processing is the creation of routes of detected targets and their localization [30,31]. The tracking stage is very often implemented using the Kalman filter algorithm. Tracking can be carried out in bistatic coordinates, as well as Cartesian coordinates (after localizing the targets). This article presents results that were obtained for a single receiving station, hence the implementation of tracking and localization algorithms was not undertaken.

## 3. 5G NR Signal Characteristics

5G-based passive radar relies on the usage of signals from 5G network traffic between gNodeB (gNB—5G NR base station) and user equipment (UE). The rules of this traffic are defined by the 5G NR standard, developed by the 3rd Generation Partnership Project (3GPP), which is described in Technical Specification, series 38 [32]. Insight into the telecommunications standard from the radar perspective is given in [9].

5G transmission is based on orthogonal frequency division multiplexing (OFDM). This means that the signal is divided into multiple data streams and transmitted simultaneously on orthogonal subcarrier frequencies. Specification [33] defines possible intervals between subcarriers, called subcarrier spacing (SCS): 15 kHz, 30 kHz, 60 kHz, 120 kHz, 240 kHz, 480 kHz and 960 kHz. In the time domain, the signal is divided into frames of 10 ms duration. Each frame consists of 10 subframes, 1 ms long. The number of slots in a

subframe varies from 1 to 64 according to Table 1. In each slot, there are always 14 symbols. An example of the frame structure is depicted in Figure 3.

**Table 1.** Number of slots per subframe.

| Subcarrier Spacing | Number of Slots |
| :---: | :---: |
| 15 kHz | 1 |
| 30 kHz | 2 |
| 60 kHz | 4 |
| 120 kHz | 8 |
| 240 kHz | 16 |
| 480 kHz | 32 |
| 960 kHz | 64 |

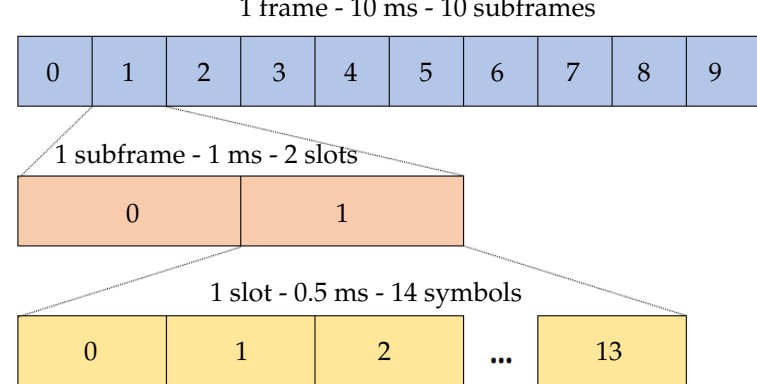

**Figure 3.** Frame structure for SCS 30 kHz.

In 5G, the smallest time interval is one symbol, and the smallest frequency quantum is a subcarrier. The resource element is the smallest time–frequency resource made up from 1 symbol and 1 subcarrier. The resource block (RB) is defined only in the frequency domain, and it consists of 12 subcarriers. All available resource blocks comprise the so-called resource grid (see Figure 4).

Resources are allocated for several signaling channels and signals, e.g., the synchronization signal block (SSB) and channel status information—reference signal (CSI-RS), but mainly for user data transmission, which is contained in the physical downlink shared channel (PDSCH) or physical uplink shared channel (PUSCH). Two directions of the traffic can be distinguished: uplink (UL)—from UE to gNB, and downlink (DL)—in the opposite direction. In most cases, the time-division duplex (TDD) is used. The time-division pattern for UL/DL transmission can be configured with up-to-symbol granularity, and the periodicity of that pattern can be defined. The DL-to-UL ratio can be any, but in practical cases most resources are allocated for downlink traffic. The work of the network scheduler is to optimally allocate the resource blocks for the signaling and data transmissions for every user in both the UL and DL directions.

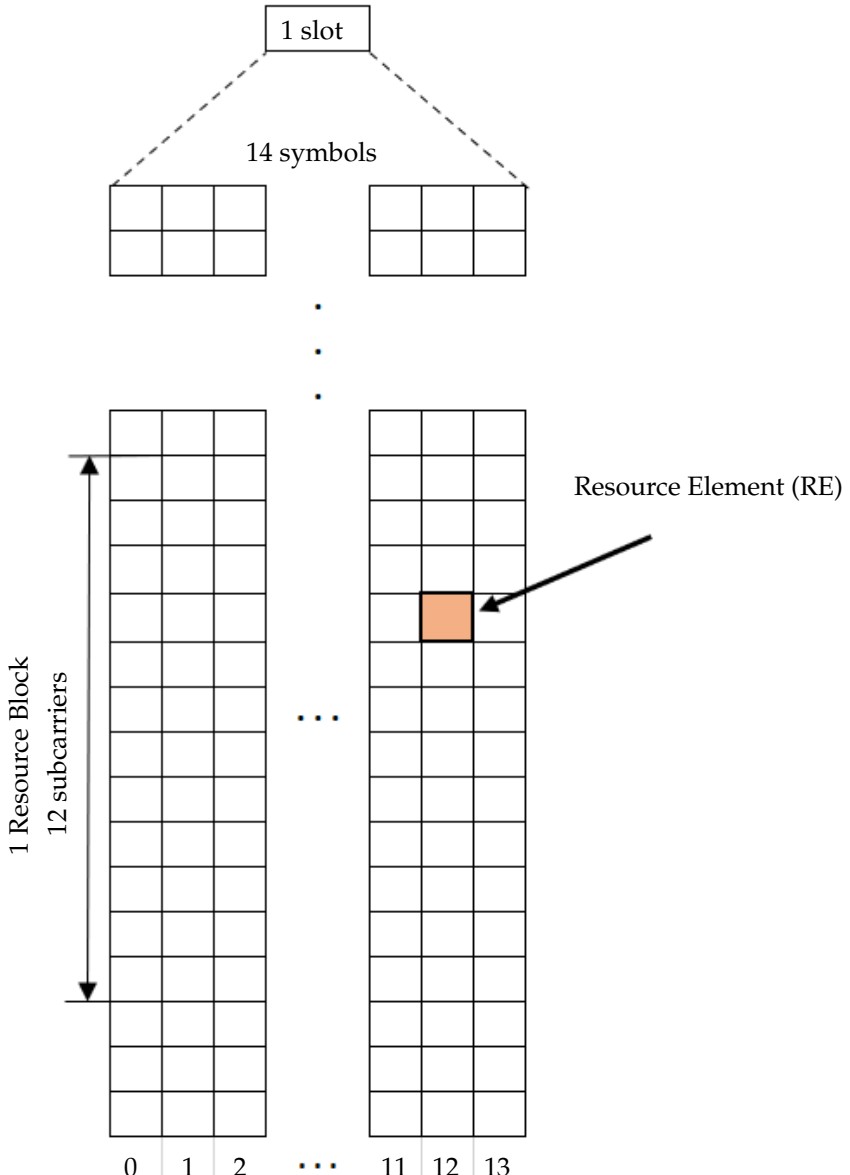

**Figure 4.** Resource grid illustration.

Theoretical cases of the allocation of the resource grid are depicted in Figures 5–7. The schemes are obtained using the 5G Waveform Generator app from Matlab. In Figure 5, the orange field is PDSCH, which is the channel responsible for the user data downlink transmission. The whole resource grid is covered with orange, which means that all possible Resource Blocks are allocated for data traffic. Purple rectangles are resources responsible for synchronization—the SSBs grouped into synchronization signal bursts (SS Bursts). For the sake of clarity, the authors did not sketch other signaling channels present in the 5G NR signal.

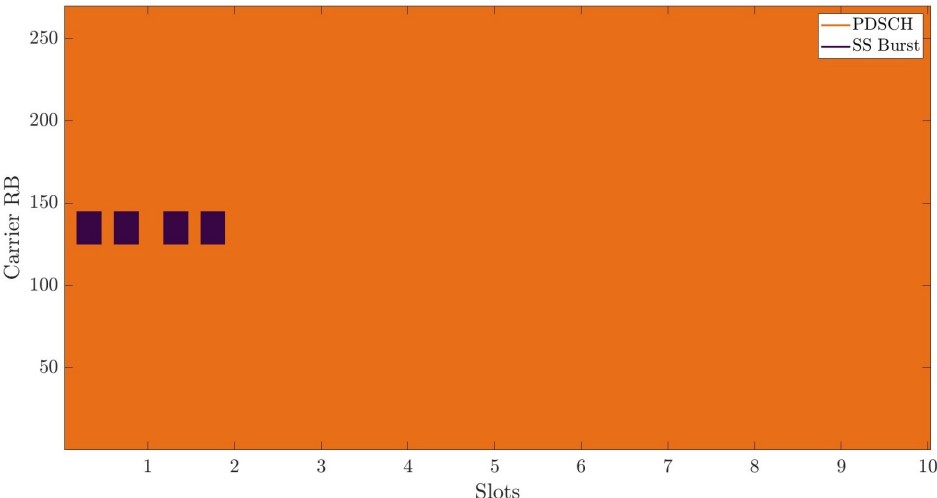

**Figure 5.** Resource grid fully allocated for data payload.

In Figure 6, there are some resources not allocated at all (white background). This is the case when there is less demand for data traffic.

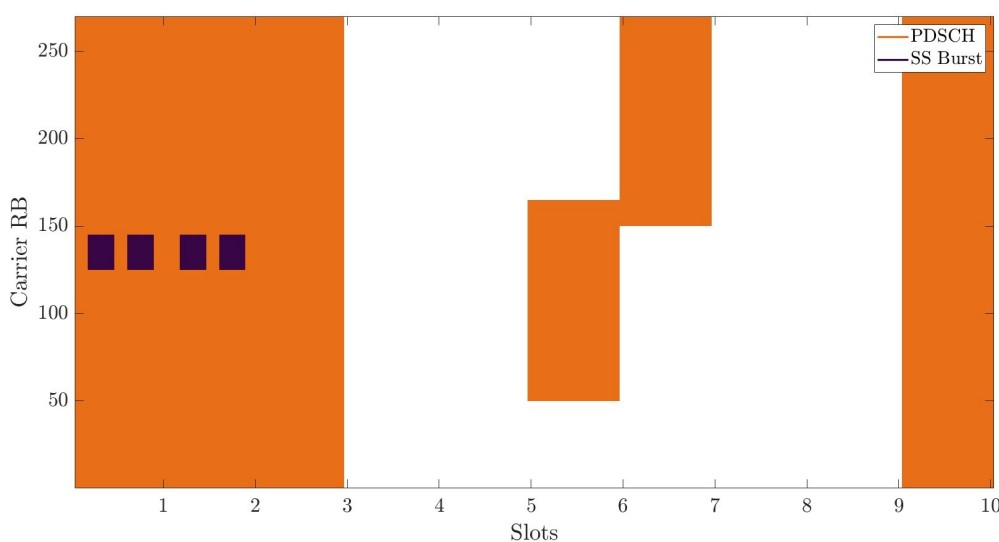

**Figure 6.** Resource grid partially allocated for data payload.

The last case (Figure 7) is when there is no data payload. The only allocated RBs are for SSB, which are "always-on" the signalling block responsible for synchronization with the base station. The rest of the time–frequency resources are not in use.

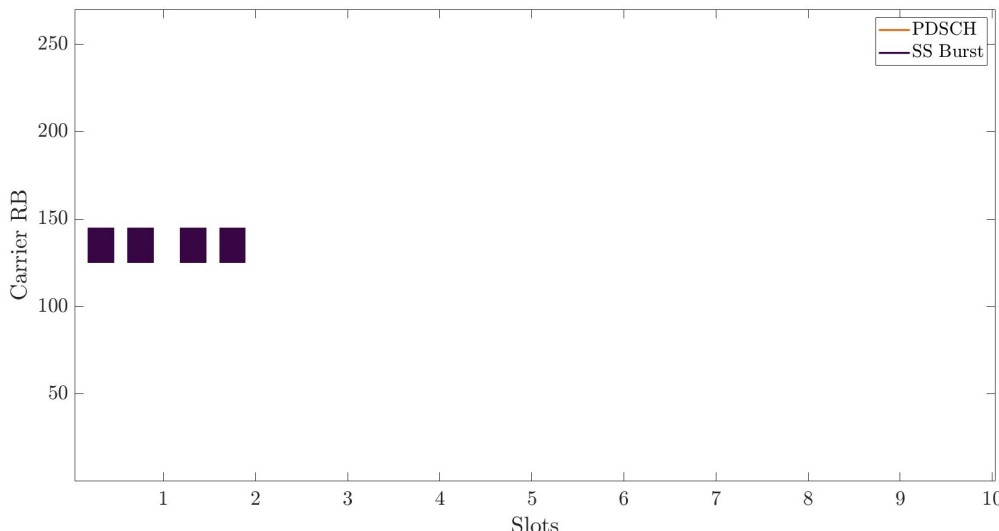

**Figure 7.** Resource grid with no data payload, only the SSB signal is present.

The above examples (Figures 5–7) show the nature of content-dependent transmission in 5G NR. When there are no active users (downloading no data), there is almost no signal transmission. As more data traffic is generated by the users, the signal becomes more filled in both the time and frequency domains. This dependency is crucial from the perspective of passive radar effectiveness.

Theoretically, both downlink and uplink transmission could be utilized for sensing [34]. Furthermore, particular signals could be considered for matched filter detections, e.g., SSB [35]. In this paper, the authors focus on passive radar based on downlink data transmission because of the relatively large bandwidth possible to be used and because of the known location of the transmitter.

The 5G NR standard covers two Frequency Ranges (FR): FR1—below 6 GHz, and FR2—mmWaves. The latter (not deployed yet) promises to unlock even larger bandwidth (up to 2 GHz), which leads to the possibility of even more precise targets' localization, imaging, or recognition using micro-Doppler features.

## 4. Practical Challenge—Content Dependency and Possible Approaches to Its Analysis

There have already been successful experiments with 5G-based radar car detection, described in [9], where the authors outline all the required processing steps. They point out the crucial issue of content-dependent transmission, but have not deeply analyzed this issue. Resource occupancy directly affects the quality or possibility of radar detection. This effect is depicted in Figure 8.

This practical challenge is even more important when considering the detection of small objects, such as drones, with a small radar cross-section (RCS). There is a clear need for a reliable method allowing the automatic selection of a time interval with sufficient signal filling.

### 4.1. Spectrum Analysis

One of the straightforward approaches for the analysis of the amount of content is the spectrum analysis of the received signal. A comparison of fast Fourier transform (FFT) plots alongside spectrograms for some content and full content is depicted in Figure 9. The FFT plots do not differ much between Figure 9b,d. Assuming thresholding as an algorithm for decision making, both cases would be classified as "content present", so the actual substantial difference of the amount of content would not be distinguished.

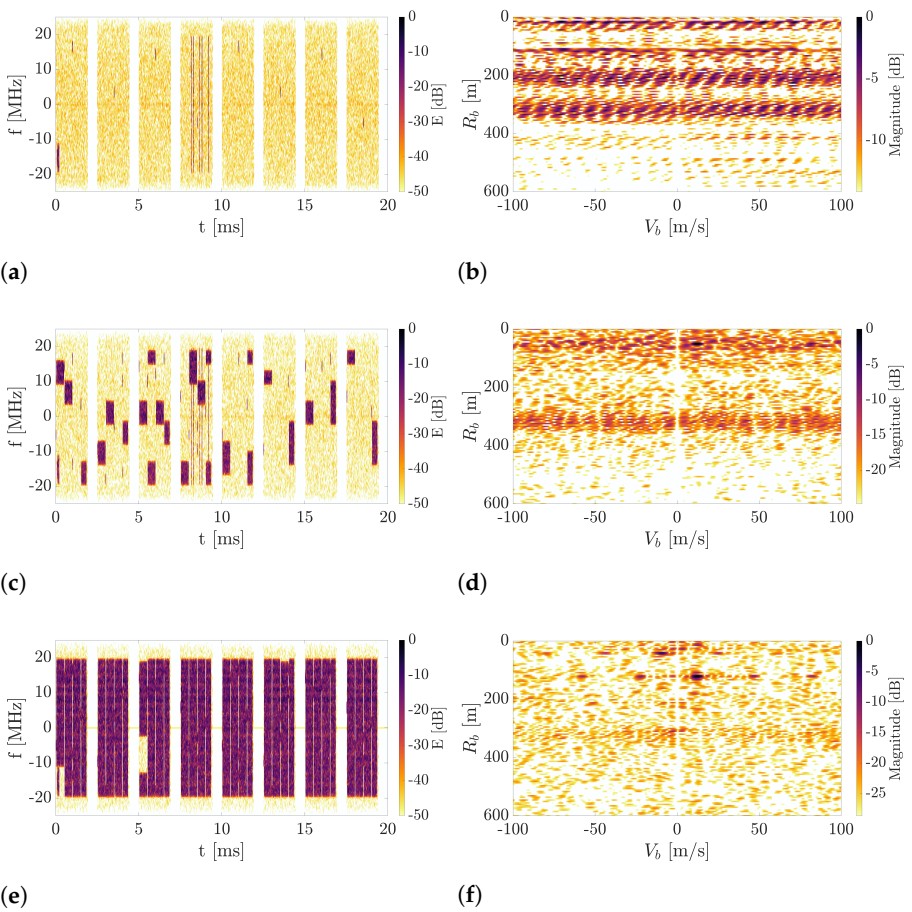

**Figure 8.** Example of spectrograms and CAFs for different amount of content in real-life 5G signal. Uplink part of the signal is blanked. (**a**) Spectrogram for no content. (**b**) CAF for no content. (**c**) Spectrogram for some amount of content. (**d**) CAF for some amount of content. (**e**) Spectrogram for almost full amount of content. (**f**) CAF for almost full amount of content.

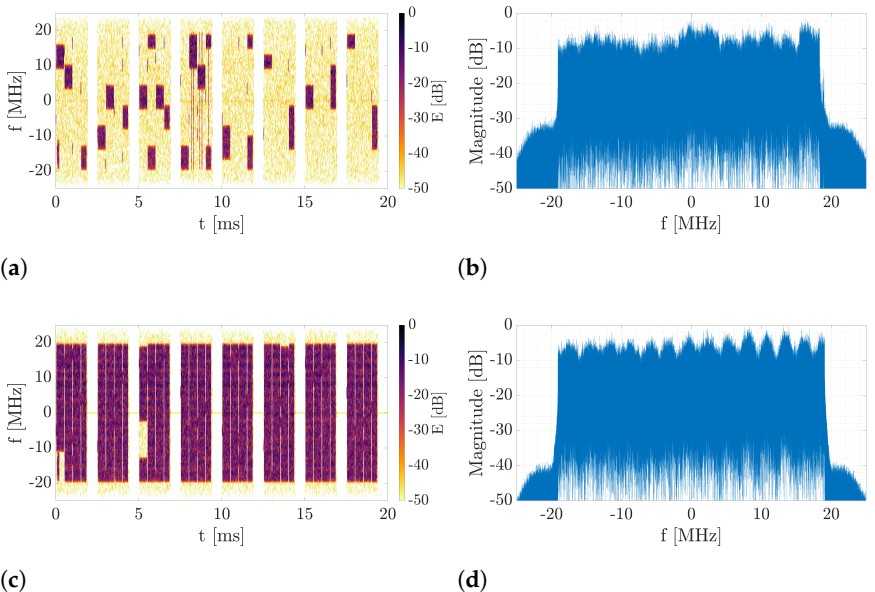

**Figure 9.** Example of spectrograms and spectrum plots for different amount of content in real-life 5G signal. (**a**) Spectrogram for some content. (**b**) spectrum for some content. (**c**) Spectrogram for no content. (**d**) spectrum for no content.

### 4.2. Power Measurement

Another basic approach is to measure the power of the received signal in a selected interval and compare it to some threshold. To evaluate this approach, 3 cases were simulated using the MATLAB computational environment and are illustrated in Figure 10. A 5G signal was generated with a duration of 20 ms. White Gaussian noise was added. The presence of the target in the 200 m range and of velocity 10 m/s was simulated. First (Figure 10a), small network traffic was simulated (10% occupancy of the resources). The measured power was 14.4 dBm. Then, (Figure 10c) the transmitted power of the data channel was scaled up, and the measured power equaled −7.9 dBm. In the last case (Figure 10e, the transmitted power of the data channel was set again as in the first case, but the occupancy of the resources was increased to 70%, so the measured power again reached approximately −7.5 dBm. Plots of the related cross-ambiguity functions show that in the first case there is no target in the range-Doppler map. In the second case, the improvement is hardly visible, as the peak at the place of the object is a few decibels above noise level. Only in the last case could the target be clearly seen, 24 dB above noise. Thresholding the measured power would select both the second and third signals for radar processing, however, only big traffic (case 3) makes the signal useful from the radar processing perspective. Thus, the trivial method based on power measurements is not sufficient.

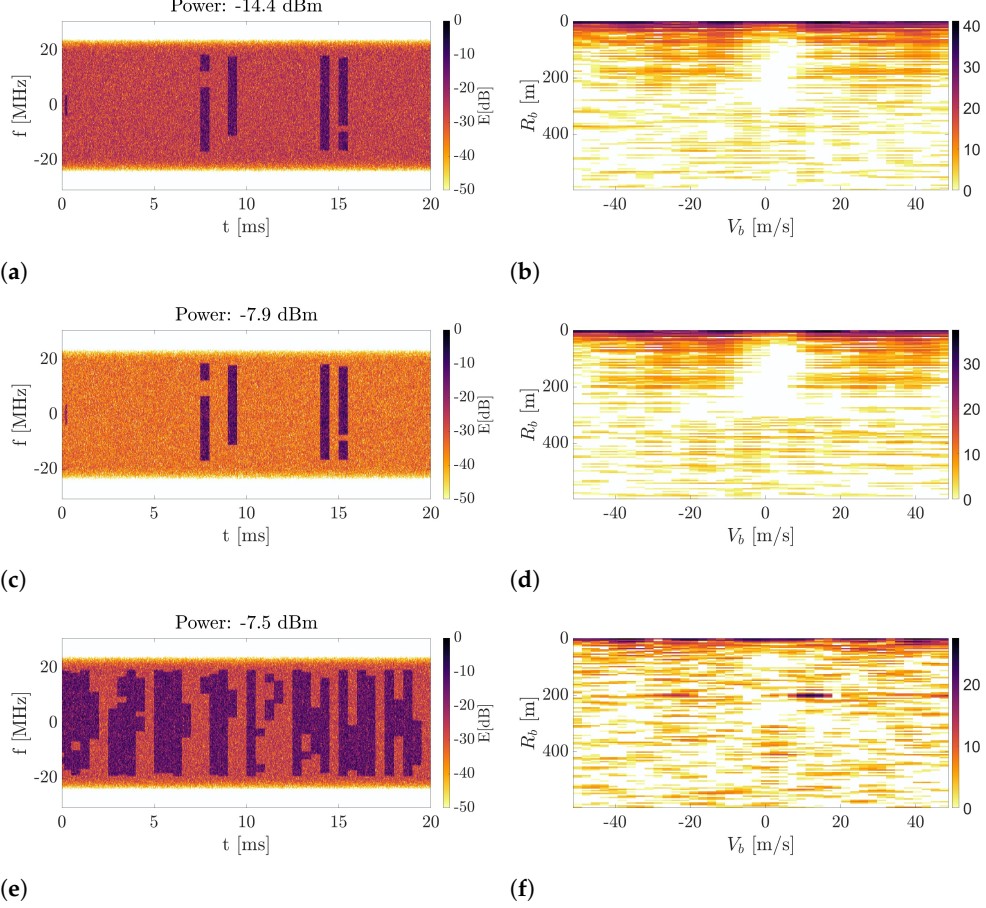

**Figure 10.** Evaluation of power measurement method. Spectrograms and CAF plots for different amount of content in real-life 5G signal. (**a**) Spectrogram for 10% of content filling. (**b**) CAF for 10% of content filling. (**c**) Spectrogram for 10% of content filling with larger transmission power. (**d**) CAF for 10% of content filling with larger transmission power. (**e**) Spectrogram for 70% content filling. (**f**) CAF for 70% content filling.

### 4.3. Root-Mean-Square Signal Bandwidth

A slightly different approach than the one presented in Section 4.1 is to determine the effective operating bandwidth based on the power spectral density (PSD) of the signal [21]. In the cited work, this method was used to analyze the root mean square (RMS) of an FM signal, which has a different nature than a 5G signal but is content-dependent as well. The calculation uses the following formula:

$$B \cdot A_{\max} = \int A(f)\mathrm{d}f, \tag{4}$$

where $B$ is the effective signal bandwidth, $A(f)$ is the PSD and $A_{\max}$ is its maximum value. Based on (4), the effective bandwidth was calculated for several fragments of real-life data, with different data link occupation. Periodograms, along with spectrograms of signal samples, are shown in Figure 11. The results presented here show that effective bandwidth can even decrease with an increase in link occupancy.

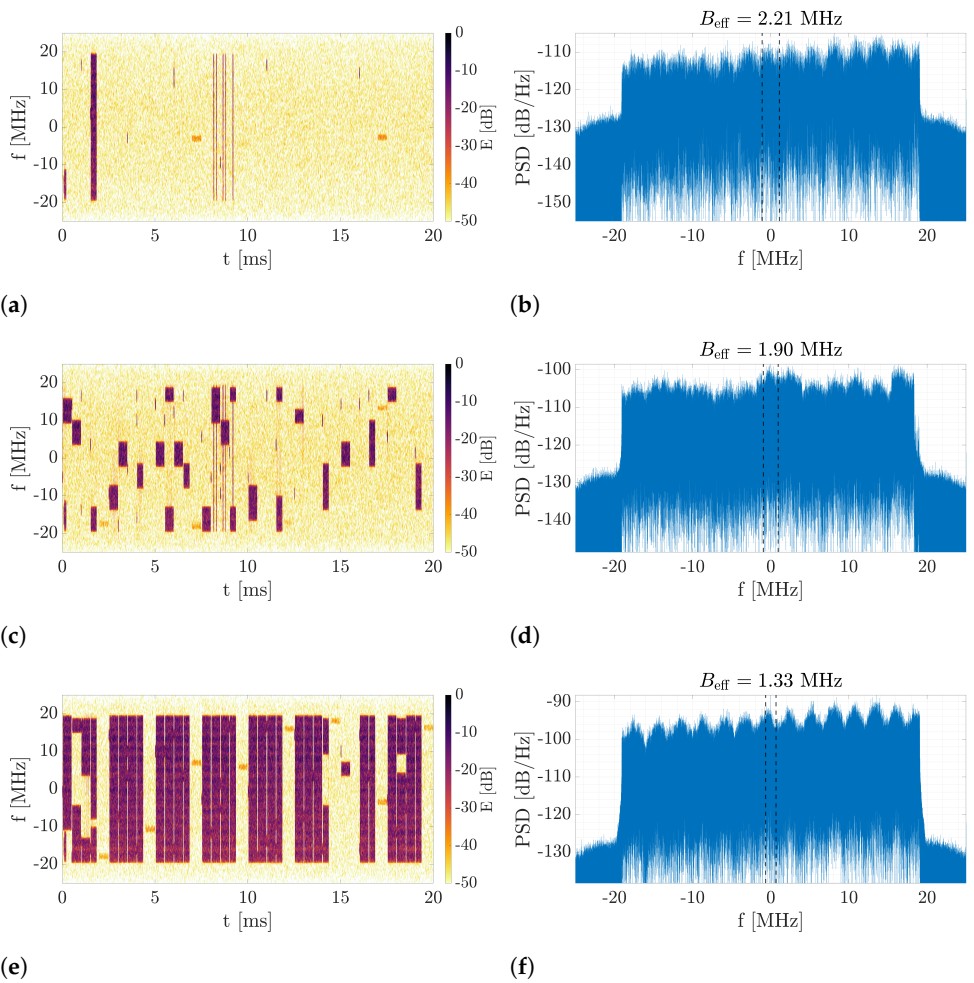

**Figure 11.** Evaluation of RMS effective bandwidth method. Spectrograms and PSD plots for different amount of content in real-life 5G signal. (**a**) Spectrogram for almost no content. (**b**) PSD for almost no content. (**c**) Spectrogram for medium amount of content. (**d**) PSD for medium amount of content. (**e**) Spectrogram for large amount of content. (**f**) PSD for large amount of content.

The bandwidth of the 5G signal, in the analyzed example, is 38.16 MHz, but the values calculated using the discussed method are much lower. A small effect of data link occupancy on the determined effective bandwidth can also be observed here. The method is sensitive to the presence of peaks and PSD distortions, hence the results obtained may be

ambiguous. Another limitation of the analyzed method is the fact that 5G signal generation is based on the OFDM technique, which in this case provides a rectangular shape of the signal spectrum envelope. This is a definite difference compared with FM modulation. Another problem in the application of this method is the large bandwidth of the 5G signal and the vulnerability of the shape of its spectrum to distortion due to radio propagation. In order to eliminate the impact of these phenomena, it would be necessary to equalize the channel in all OFDM symbols, which is impossible to implement in practice. Hence, the possibility of using a method based on estimating the effective bandwidth of the signal to select time moments for PCL processing is marginal.

## 5. Rényi Entropy for 5G Signal Analysis

### 5.1. Theory

Let us assume an OFDM signal model given as follows [36]:

$$x(t) = \frac{1}{\sqrt{N}} \sum_{k=0}^{N-1} d_k e^{j2\pi f_k t}, \quad 0 < t < T, \tag{5}$$

where $f_k$ is the frequency of the $k$th subcarrier (from the set of $N$ orthogonal subcarriers), $d_k$ is the $k$th complex data symbol and $T$ is the duration of the OFDM symbol. Equation (5) is a general formula defining signal performance depending on subcarriers. The model is valid for 5G signals and describes the inverse Fourier transform on modulated subcarriers. From (5), it is clear that the higher the number of subcarriers, the more complex the signal character is. In 5G networks, the number of subcarriers and their duration change continuously depending on the transmitted content. The distribution of subcarriers in a time–frequency grid links the amount of data to the possible resources of the system. A time-domain signal is obtained by performing the inverse Fourier transform, ultimately used for data transmission. The OFDM signal reaching the radar receiver can easily be transformed into a time–frequency grid by performing a short-time Fourier transform (STFT). This makes it possible to directly analyze the allocation of subcarriers in the network and the use of time resources without knowing the signal character and network configuration. The STFT of the complex and continuous signal $x(t)$ and the even and real window $h(t)$ is defined as [37]

$$F_x^h(t, \omega) = \int_{\mathbb{R}} x(\tau) h^*(\tau - t) e^{-j\omega(\tau - t)} d\tau. \tag{6}$$

The energy distribution, usually referred to as a spectrogram, is given as

$$S_x^h(t, \omega) = |F_x^h(t, \omega)|^2. \tag{7}$$

To express the amount of information contained in the STFT, the Rényi entropy is usually applied. The Rényi entropy was proposed in the context of measuring the content contained in the time–frequency representation [38]. It is very often used as a measure of a method's ability to concentrate distribution, e.g., for reassignment or synchrosqueezing techniques [39]. In this work, Rényi entropy is used in a different context: namely, it serves as a quantitative measure of resource utilization in a 5G network. The $\gamma$-order Rényi entropy of any time–frequency signal representation $\text{TF}_x(t, \omega)$ is expressed in bits (for simplicity, the results given in the further part of this work are presented without a unit) and defined as [38]

$$H_R^\gamma(\text{TF}_x) = \frac{1}{1 - \gamma} \log_2 \left( \frac{\iint_{\mathbb{R}^2} (\text{TF}_x(t, \omega))^\gamma dt d\omega}{\iint_{\mathbb{R}^2} (\text{TF}_x(t, \omega)) dt d\omega} \right). \tag{8}$$

In this work, the Rényi entropy is computed using the absolute value of the STFT, such that $|F_x^h(t, \omega)|$, and its order was fixed to $\gamma = 3$ for all experiments. As shown in the literature [38,40–43], $\gamma = 3$ ensures stable results and is the best choice for information analysis. Thus, further considerations are carried out within this assumption. Since the

signal given by (5) is an inverse Fourier transform on the modulated subcarriers, the Fourier transform performed on this signal allows the observation of variability in the number of subcarriers. Thus, it is possible to give a quantitative relationship informing the utilization of downlink resources using the Rényi entropy. An illustrative example of the Rényi entropy with different values of the downlink allocation and noise level is shown in Figure 12. In the experiment, the signal was simulated in the MATLAB environment with an assumed portion of subcarriers corresponding to the percentage of allocated resources. The results clearly show a strong relationship between the transmitted content and the entropy value.

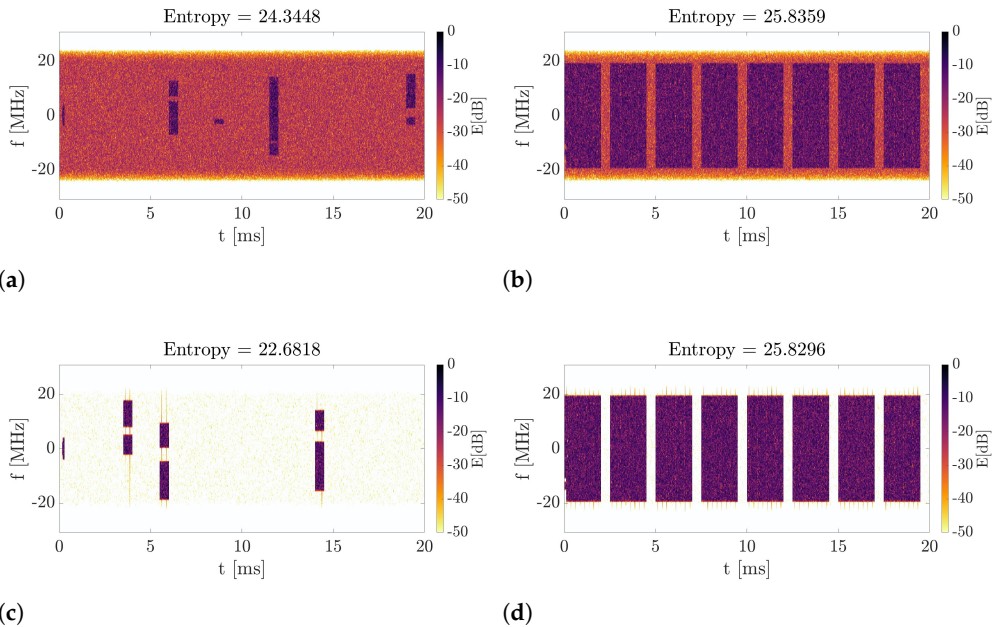

**Figure 12.** A comparison of the Rényi entropy value for different filling and SNR. (**a**) Spectrogram of the simulated 5G transmission with 5% allocated subcarriers and SNR = 10 dB. (**b**) Spectrogram of the simulated 5G transmission with 100% allocated subcarriers and SNR = 10 dB. (**c**) Spectrogram of the simulated 5G transmission with 5% allocated subcarriers and SNR = 40 dB. (**d**) Spectrogram of the simulated 5G transmission with 100% allocated subcarriers and SNR = 40 dB.

An additional experiment showing a direct dependence between the entropy and a signal filling is depicted in Figure 13. For five noise values SNR = $\{0, 10, 20, 30, 40\}$ dB, 5G signals were generated with different resource allocation ranging from 0 to 100 %. A one-hundredfold Rényi entropy was calculated for each frame fill value and averaged. The results show that a direct dependence between the entropy and downlink resource allocation exists that can be used to make an adaptive signal integration algorithm in passive radar. Then, regardless of the power or frequency of the signal, one can calibrate the system, e.g., by burdening the 5G network. The base station will then generate a full downlink transmission, allowing the maximum entropy value to be determined. After that, the passive radar can perform adaptive integration only if the amount of data transferred is sufficient to achieve the required resolution.

Furthermore, the entropy values for cases presented in Section 4 were obtained. Entropy for a "no content" case equals 19.63, for "some amount of content" 23.41 and for "full content" 24.56. This proves that the entropy-based method outperforms the simple spectrum analysis method (Figure 9), as well as effective bandwidth measurement (Figure 11), in which the difference in signal filling could not be distinguished. The results from the power measurement example (Figure 10 are as follows: 10% of content and low power: 24.56, 10% of content and high power: 23.75 and 70% with low power: 25.72. Furthermore, in this case, the entropy-based approach shows its reliability, which is unobtainable in the power measurement method.

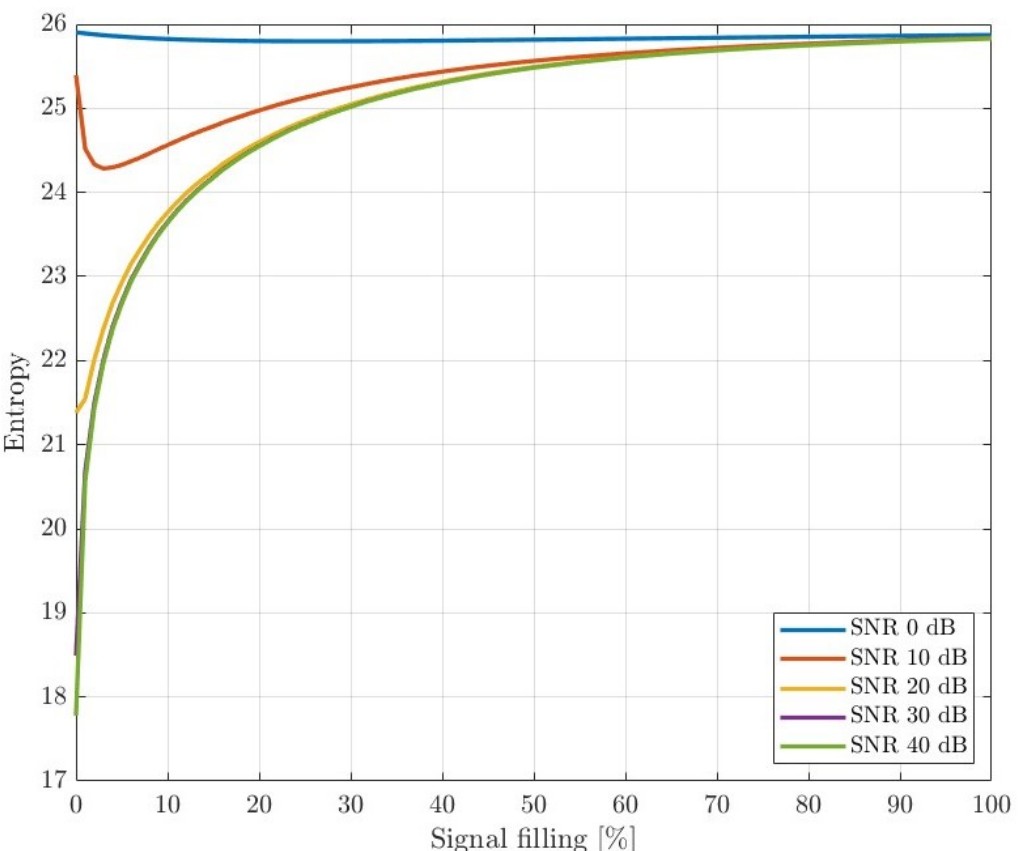

**Figure 13.** Plot entropy—resource usage.

*5.2. Rationale Behind the Use of the Rényi Entropy for Drone Detection*

Let us consider the passive radar range Equation [44]

$$R_e < \sqrt{\frac{P_T G_T G_R S_0 \lambda^2 T_{\text{int}}}{(4\pi)^3 L_0 D_0 k T_0}}, \tag{9}$$

where $R_e = \sqrt{R_1 R_2}$ is the equivalent of the monostatic detection range ($R_1$ is the range from the transmitter to the target and $R_2$ is the range from the target to the receiver), $P_T$ is the transmitter power, $G_T$ and $G_R$ are the transmit and receive antenna gain, respectively, $S_0$ is the target's RCS, $L_0$ stands for the total propagation and system losses, $D_0$ is the detection factor, $k$ is the Boltzmann constant equal to $1.38 \times 10^{-23}$ J/K and $T_0$ is the effective noise temperature of the receiver, which depends on its noise figure, according to Equation (10) [23]:

$$T_0 = T_{\text{ref}}(10^{N_f/10} - 1), \tag{10}$$

where $N_f$ stands for noise figure and $T_{\text{ref}}$ is reference temperature, usually assumed to equal 290 K. Most of these parameters are related to a particular system and the passive radar receiver, e.g., wavelength and antenna gains. The most variable parameters that are not related to the equipment belonging to the transmitting station and passive radar are the integration time $T_{\text{int}}$ and the RCS $S_0$. Drones are usually considered as low-RCS targets due to their small size and the material they are made of. Therefore, the detection of such targets requires an extension of the integration time to increase the SNR as much as possible.

In 5G networks, a signal duty factor (resource allocation) depends on the data transmitted to terminals. If few data are transferred, only part of the time and frequency resources

are used. This results in an artificial reduction in the integration time because it may turn out that for the given $T_{int}$, the signal was present, for example, 10% of the time. This leads to a decrease in the SNR in the receiver and deterioration of the range resolution. Therefore, autonomous selection of the integration time to optimize the detection probability is essential. Thus, this article uses Rényi entropy to automatically select a signal that allows optimization of the radar equation, i.e., a signal that gives the highest possible effective use of the integration time. Since the Rényi entropy can capture the inner properties of the OFDM signal used in 5G networks, its computation determines the temporal possibility of target detection. Indeed, if the entropy increases, the number of subcarriers (bandwidth) and their duration also increases. Therefore, the effective integration time becomes higher, enabling the detection of a target located far from the passive radar receiver (the right-hand side of (9) grows). In systems characterized by transmitting a signal with variable time and bandwidth, the value of the $T_{int}$ parameter differs from the actual integration time. Considering how much of the integration time was effectively taken up by the signal, the actual radar detection range may be much smaller, which results from the theory (9). In order to inspect the signal properties in terms of resolution and coverage, the authors of this article use Rényi entropy, the values of which are related to the time and bandwidth used by the 5G network during data transmission from a non-cooperating base station. This statement can be supported by a simulation analysis, described in Section 6.

In a practical application, the proposed technique can be used comprising the following steps:

1.  Calibration (if possible)—record the 5G signal with the maximum allocation of resources. In civilian systems, this is easy to meet and comes down to the load on the network by downloading large amounts of data (e.g., large files) using one or more 5G mobile terminals. As a result, the base station will use the possible resources allowing the maximum entropy value to be assessed. In the case where calibration with the use of terminals is not possible, one should constantly verify the received signal and analyze the maximum value of entropy on an ongoing basis, indicating the use of a large number of resources.
2.  Signal reception—the signal is received continuously in the same way as a typical passive radar.
3.  Useful signal extraction—the signal is analyzed in terms of resource allocation by the 5G network. Only those fragments of the signal that meet the adopted condition for the Rényi entropy level are selected (e.g., signal frames with a filling exceeding 90% of the maximum entropy value).
4.  Passive radar processing—classical target detection, as shown in Figure 2.

## 6. Simulations

Simulations were carried out to prove the importance of selecting a long enough time interval for radar detection of drones and evaluate the aforementioned entropy-based method of measuring the content contained in 5G NR signals.

The simulation parameters are listed in Table 2 and were chosen to coincide with real-life experiments described in the further part of this article.

First, a simple experiment was carried out taking into account the relationship defined by (9). For different values of the integration time $T_{int}$ and the radar cross-section $S_0$, the detection range was calculated. The results are illustrated in Figure 14. As can be seen, the detection range strongly depends on $T_{int}$. Thus, increasing the integration time by adaptive signal selection increases the probability of detection as well.

**Table 2.** Simulated network parameters.

| Name of the Parameter | Value |
|---|---|
| Center Frequency | 3.44 GHz |
| EIRP ($P_t G_t$) | 73 dBm |
| Receiver antenna gain $G_r$ | 10 dBi |
| Threshold $D_0$ | 11 dB |
| Integration time $T_{\text{int}}$ | 20 ms |
| Total losses $L_0$ | 10 dB |
| Effective noise temperature of the receiver $T_0$ | 493 K |
| Radar cross-section $S_0$ | 1, 10, 50, 100 m$^2$ |

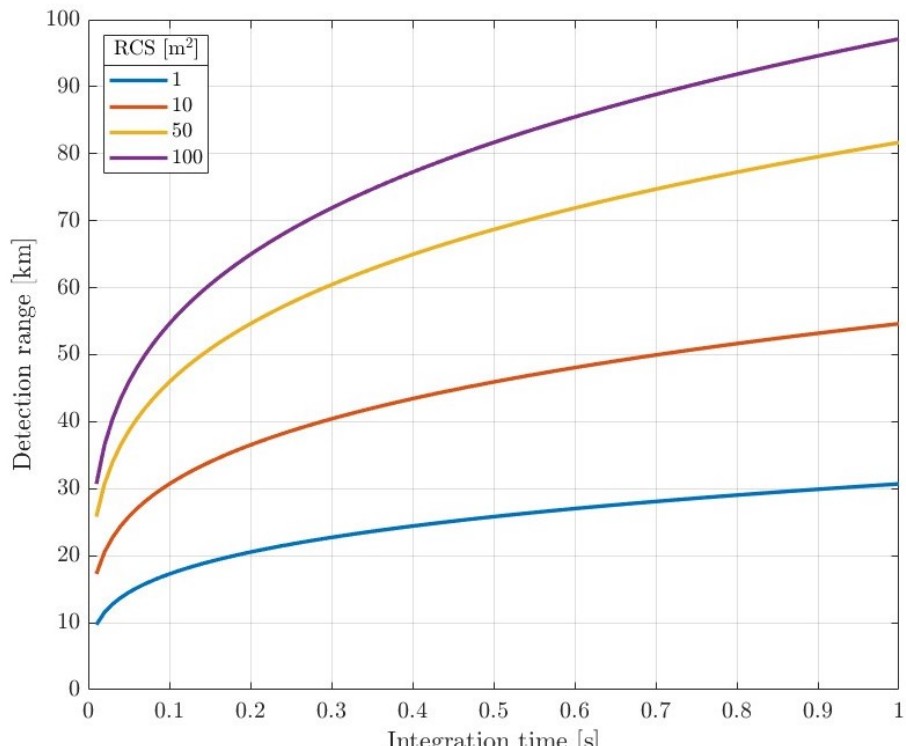

**Figure 14.** Plot range—integration time.

Secondly, synthetic signals compliant with the 5G NR standard were generated. The filling varied from 0 to 100% of the downlink channel slots, with random positions in the time–frequency allocation grid. The signal parameters were set as follows: channel bandwidth: 40 MHz, sampling rate: 61.44 MHz and subcarrier spacing = 30 kHz. A time delay and Doppler shift were added to simulate the reflection of a single moving target. White Gaussian noise was added with a power level referring to SNR values of 0, 10, 20, 30, and 40 dB. The time interval for each calculation was 20 ms. Then, the average value of the Rényi entropy for each filling and SNRs was obtained. Results are presented in Figure 13. It can be seen that the value of the entropy increases with signal filling.

The same simulated echo signals were applied to measure the probability of detection ($P_d$) in 5G-based passive radar as a function of signal filling. Signals were processed according to [9] with the following steps: uplink cancellation, signal filtering and clutter removal, CAF calculation and target detection (using classical CFAR algorithm). Results are

presented in Figures 15–17 for different probabilities of false alarm $P_{fa} = 10^{-4}$, $P_{fa} = 10^{-6}$, and $P_{fa} = 10^{-8}$, respectively.

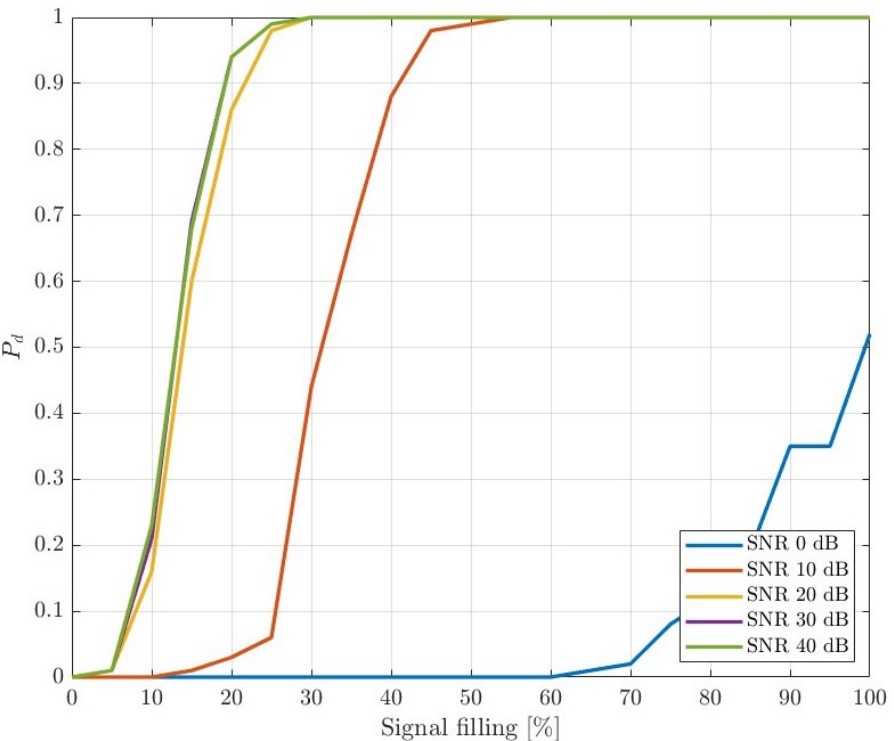

**Figure 15.** Probability of detection ($P_d$) as a function of signal filling ($P_{fa} = 10^{-4}$).

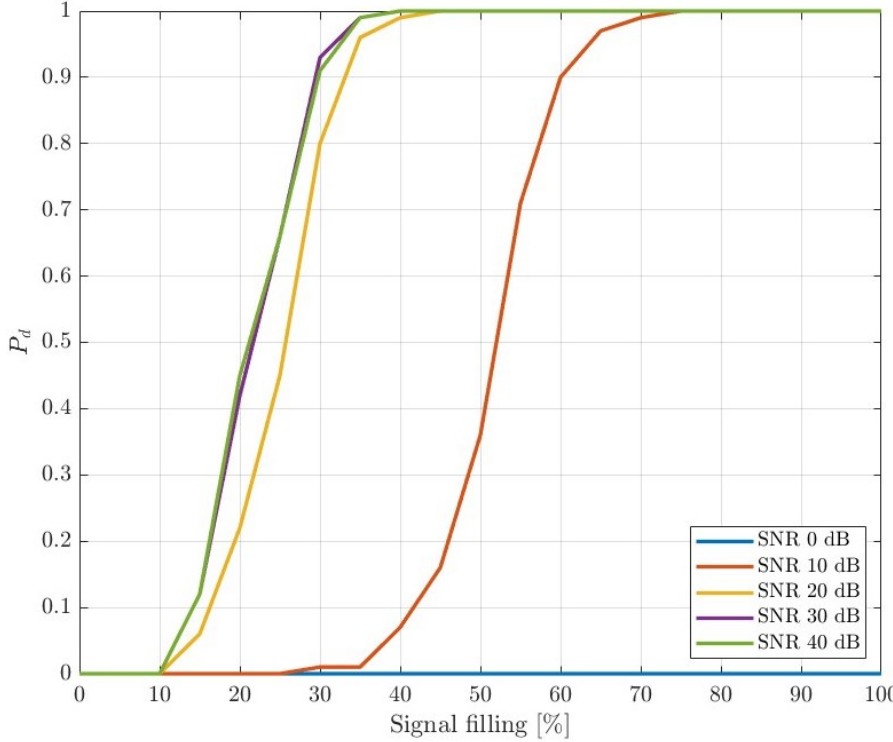

**Figure 16.** Probability of detection as a function of signal filling, $P_{fa} = 10^{-6}$.

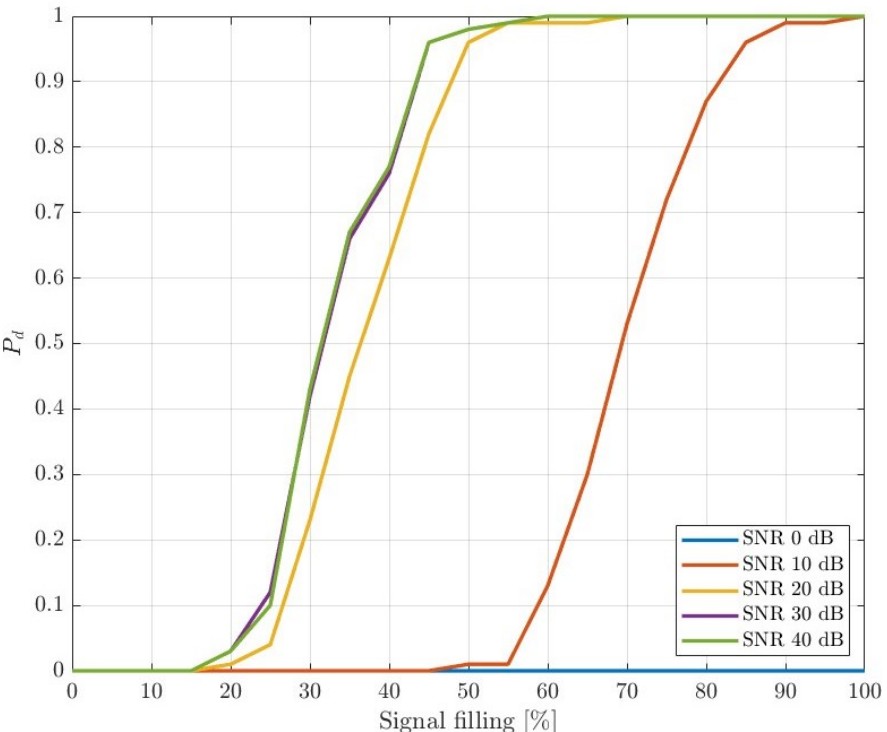

**Figure 17.** Probability of detection as a function of signal filling, $P_{\text{fa}} = 10^{-8}$.

The results clearly show a leverage of the signal filling on the possibility of target detection. In line with expectations, the higher the resource allocation, the better the detection performance. Furthermore, the differences between high SNRs (especially those greater than 20 dB) are marginal, and one can assume that such values can be relatively easily achieved in short-range 5G-based PCL applications. The plots also show the obvious dependence of the detection probability $P_{\text{d}}$ on the assumed $P_{\text{fa}}$.

Simulations showed the importance of sufficient integration time and resource allocation, especially for low-RCS target detection. It also evaluated the Rényi entropy as the right measure of signal resource allocation.

## 7. Drone Detection

### 7.1. Experiment Description

The experiment was carried out at the campus of the Technical University of Łódź, Poland, owned by a cooperative 5G network used as a source of illumination for passive radar. The main idea of the measurement was to record the signal reflected from a cooperative target in various scenarios possible in a 5G-network-based passive radar. The measurement setup and target are shown in Figure 18. The target of interest was a drone (DJI M600 PRO), illustrated in Figure 18b. Its size was ca. 1 m in width and length, ca. 40 cm in height and was made of plastic. The target's RCS, for the operating frequency of about 3.44 GHz, is not known and needs to be characterized [45]. For measurements, the authors used a dual-channel radar receiver whose antennas (connected to corresponding channels of the recorder) were mounted as in Figure 18b. The reference antenna was pointed towards the 5G base station. The second antenna was observing the illuminated space where the drone was flying, as shown in Figure 18c. The surveillance antenna was hidden behind the wall to minimize the influence of the reference signal in the surveillance channel (DPI leakage). As shown in Figure 19, the surveillance channel was composed of a relatively high gain amplifier with a gain above 20 dB. Thus, the antenna position was dictated to avoid the signal saturation resulting from the presence of the transmitter at a relatively low range. The reference channel has a gain of ca. 10 dB. Both signals are

passed through to the SDR based on the Ettus USRP X310, digitized and stored on the hard drive. To ensure frequency synchronization, the GPS signal is used. Data storage is carried out using a mid-class computer with an Intel Core i7-9700 2.66 GHz processor, 32GB RAM, SSD drive (separate drive for each channel) and the Ubuntu operating system. The data were processed off-line after signal reception. The drone was recorded during several maneuvers, and the selected results are illustrated in the further part of this section.

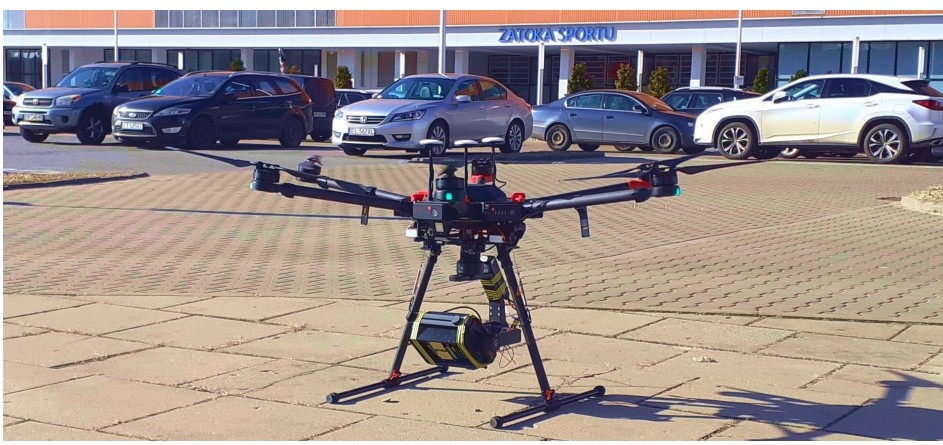

(**a**)

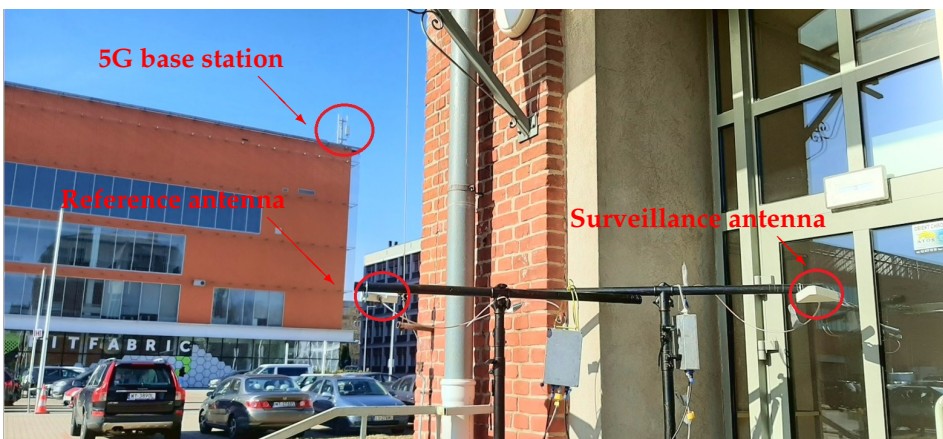

(**b**)

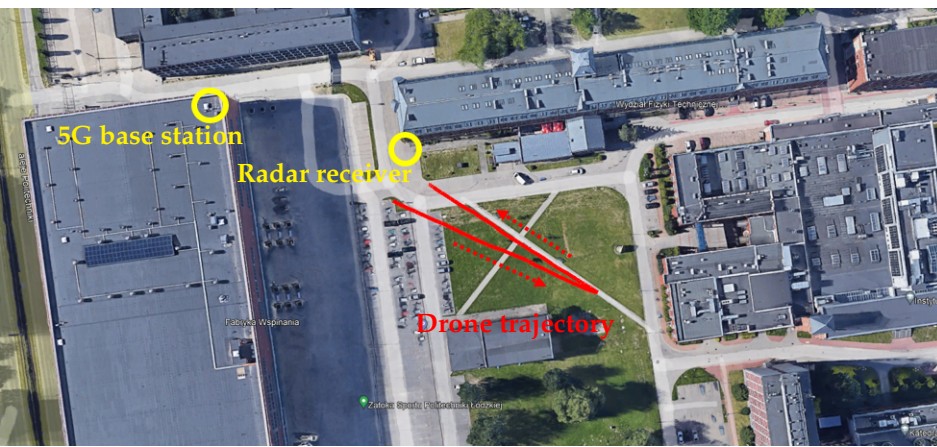

(**c**)

**Figure 18.** Drone and the recorder during the trials with the drone trajectory superimposed on the map. (**a**) Cooperative target. (**b**) Measurement setup. (**c**) GPS trajectory of the drone. The red dashed arrows indicate the drone's direction of flight.

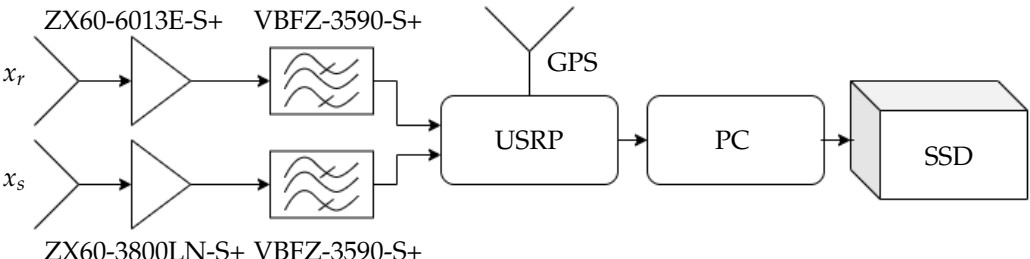

**Figure 19.** Recorder diagram.

## 7.2. Results

First, like in the simulations, the part of the signal responsible for uplink transmission was deleted. Then, the process of calibration was conducted. It consisted of recording the reference signal with 100% of allocated resources. Then, the Rényi entropy was calculated. The reference maximum value was 25.67 (see Figure 20). For further analysis, it was assumed that the threshold above which the signal is suitable for radar processing is 25.5.

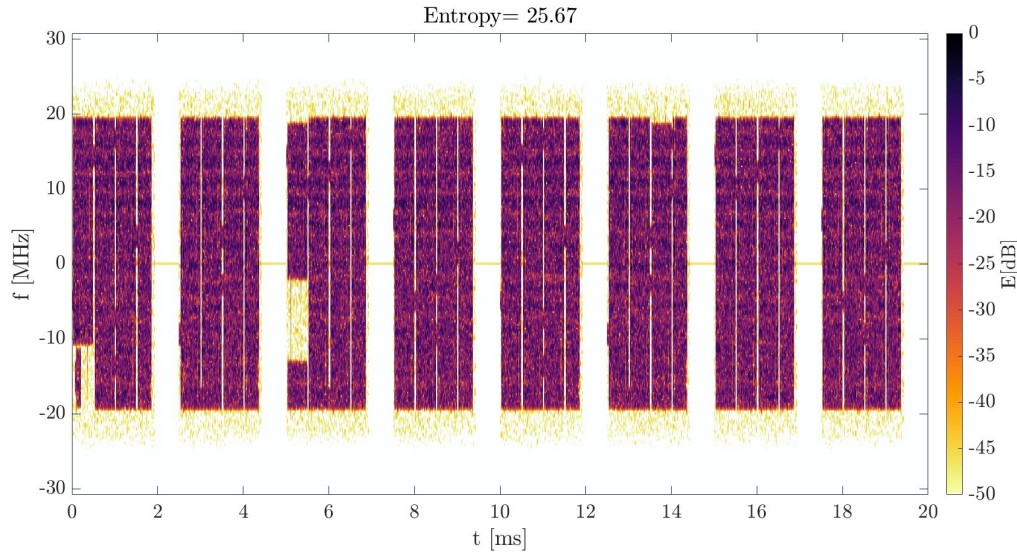

**Figure 20.** Spectrogram of the reference signal with maximum resource allocation.

Using this value, the time interval for the detection was chosen. The cross-ambiguity function where the drone position could be clearly seen is depicted in Figure 21. Its bistatic range and velocity are compatible with data from GPS logs. In this case, integration time equals 20 ms.

Measurement of the entropy allowed for the discovery that there were longer periods with enough content in the signal. Therefore, the integration time was successfully increased to the value of 100 ms without loss of the target's detection capabilities. This leads to better velocity resolution. The results are shown in Figure 22.

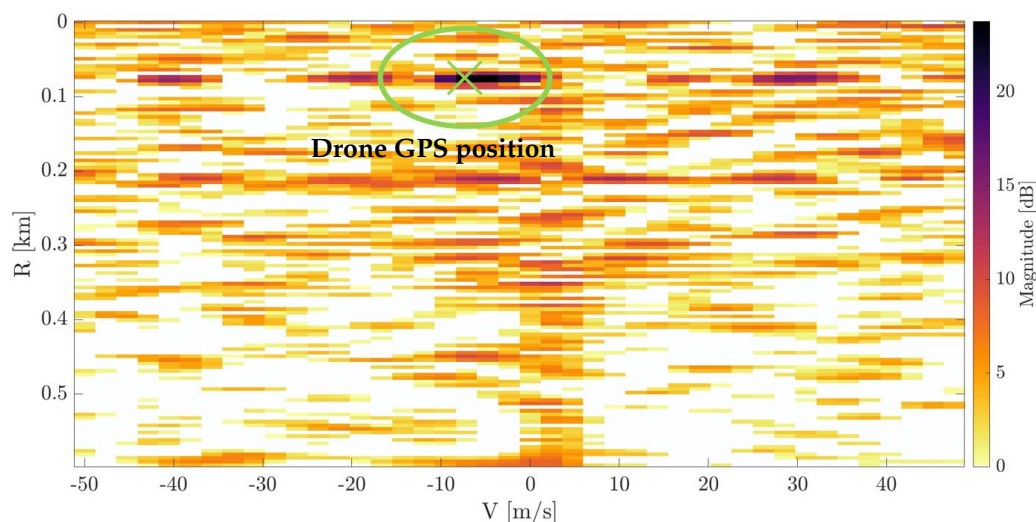

**Figure 21.** CAF with drone detection ($T_{\text{int}} = 20$ ms) with reference to its GPS position (green cross).

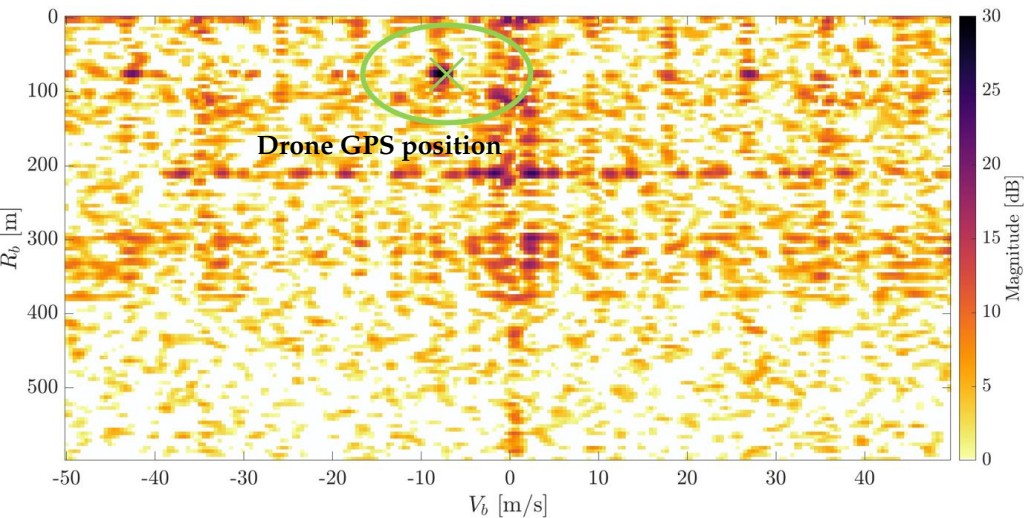

**Figure 22.** CAF with the drone detection ($T_{\text{int}} = 100$ ms) with reference to its GPS position (green cross).

Then, the 1-minute flight of the drone was analyzed. The trajectory of the flight is shown in Figure 18c. At the time when the Rényi entropy values were above the threshold, the radar processing was conducted. On the calculated CAFs, the CFAR detection was made. Results are shown in Figure 23. With a black line, the bistatic range and velocity from the GPS logger is drawn for reference. The red dots are the cumulative history of bistatic radar detections obtained. Integration time was set to 100 ms. This value provides sufficient velocity resolution and a convenient refresh rate for detection plots for the purposes of target tracking. At the same time, 100 ms of integration time gives a higher probability of finding the signal fragment with a sufficient Rényi entropy value. The SNR threshold was set to 15 dB.

The proposed method allows for successful drone detection for almost the entire flight path. Thanks to the selection of those frames with a dense resource allocation, the authors confirmed the possibility of detecting UAVs in 5G-based short-range PCL applications. This would be impossible with the technique from the literature [21], since the method is not viable in dealing with the nature of 5G signals, which can be seen in Figure 24. It shows the obtained values of the Rényi entropy and the effective bandwidth computed according to [21] for 20 ms fragments across the whole 60 s signal. Values of the entropy indicate the full allocation is stable in the whole selected interval, while the effective

bandwidth plot shows great variability. The entropy value of approximately 25.7 shows the full resource allocation, which was constituted during the system calibration (see Figure 20). The estimated bandwidth resulting from the use of the reference method is unstable and does not cover the actual signal properties. Thus, the experiment confirmed the performance of the proposed method with real-life signals and the efficacy of drone detection using the new illuminating waveform.

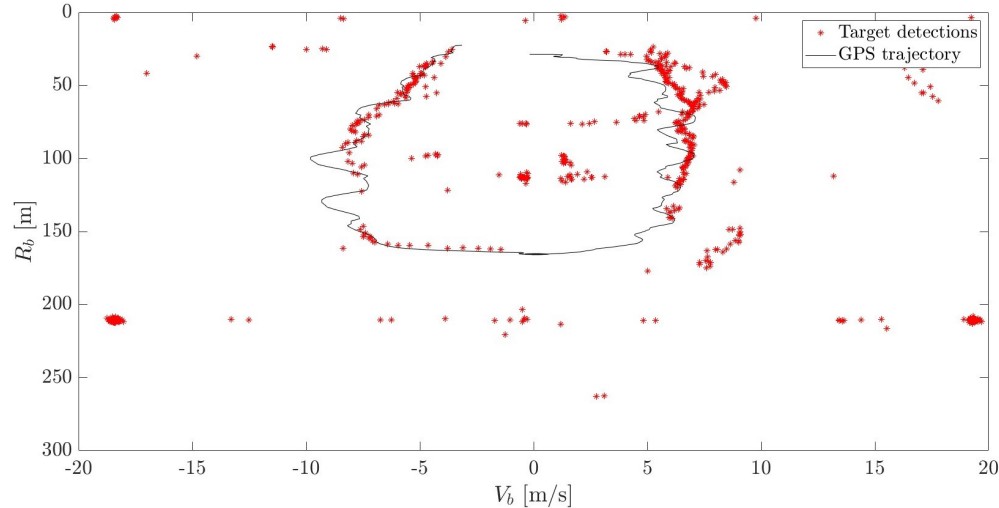

**Figure 23.** Detection of 1 minute of the drone flight.

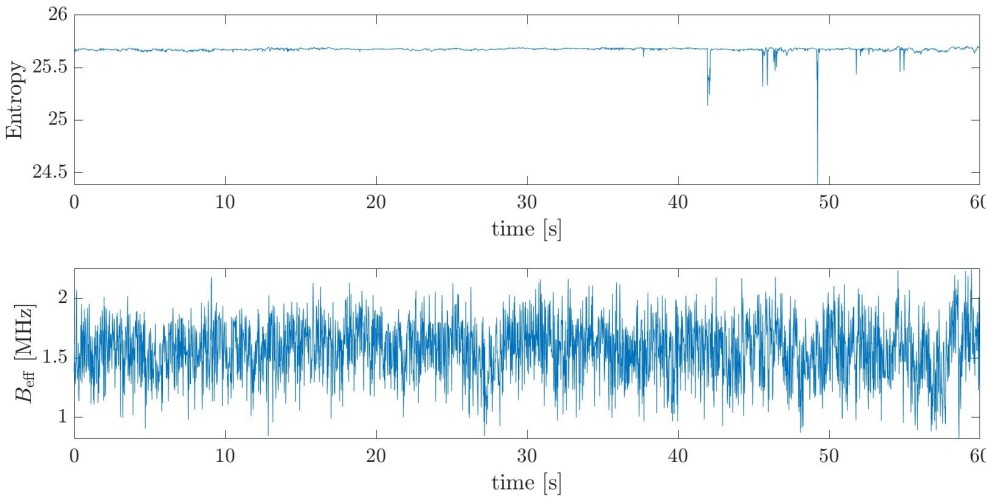

**Figure 24.** Entropy and effective bandwidth values obtained for consecutive 20 ms fragments across 60 s of signal.

## 8. Conclusions

In this paper, the successful experiment of drone detection using 5G-based passive radar was shown. The target was a medium-sized drone with a GPS logger used as a reference position. For the first time in the literature, the successful detection of a 1-min flight was shown. To the authors' best knowledge, there are no previous results of such an experiment published. The issue of content dependency was addressed. Some methods to overcome it were evaluated as not sufficient and a new approach, employing the Rényi entropy, was proposed. Simulations were conducted to prove this concept. It was shown that the evaluation of the Rényi entropy of a 5G signal enables a reliable decision on the time moment and its duration for radar processing, which is especially important in small-target detection such as drones. It must also be highlighted that the proposed method can

be widely used in content-dependent wireless transmissions, e.g., LTE, used for passive radar purposes. As shown in the paper, the proposed technique for adaptive integration is superior to the technique from the literature. Thanks to dependable and autonomous operation, the proposed method is suitable for operational PCL systems. However, the proposed approach is functional as long as the content is present in the downlink transmission. Otherwise, the 5G network only generates synchronization pulses with a relatively low repetition rate (from 5 to 160 ms), which entails significant limitations [35].

Additional works may involve real-time implementation of the proposed processing. Moreover, for continuous resource allocation, one can extract micro-Doppler signatures and estimate target parameters. A promising concept is to apply the proposed technique to detect smaller targets such as birds and tiny drones.

**Author Contributions:** Conceptualization, R.M. and K.A.; Methodology, K.A., P.S. and M.P.; Software, R.M., K.A. and M.P.; Validation, K.A., P.S. and M.P.; Formal analysis, K.A.; Investigation, R.M. and M.P.; Writing—original draft, R.M., K.A., P.S. and M.P.; Visualization, R.M., K.A. and M.P.; Supervision, P.S.; Project administration, R.M. and P.S. All authors have read and agreed to the published version of the manuscript.

**Funding:** This research received no external funding.

**Data Availability Statement:** Not applicable.

**Acknowledgments:** The authors would like to thank Sławomir Hausman and Piotr Korbel for providing access to the 5G infrastructure at the Łódź University of Technology and for their valuable help during the measurement campaigns, as well as Krzysztof Kulpa for their support during the trials. The authors would like to also express their deepest gratitude to Tomasz Zieliński, Jacek Wszołek and Adam Księżyk for their significant contribution to understanding 5G signals.

**Conflicts of Interest:** The authors declare no conflict of interest.

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
