# Peer review of "Rényi Entropy-Based Adaptive Integration Method for 5G-Based Passive Radar Drone Detection"

_remotesensing, doi:10.3390/rs14236146_

Round 1
Reviewer 1 Report
Review report 1:Major revision
Comments and Suggestions for Authors
In this paper, authors introduce a novel Rényi entropy based adaptive strategy for signal integration of the 5G-network-based passive radar system. The paper explains the 5G passive radar principles, 5G NR signal characteristics and the practical challenges in detail. And then the authors developed the Renyi entropy to automatically select a signal that allows optimization of the target detection. Simulation and actual measurement prove the effectiveness of the scheme. Finally, the paper concludes that proposed method can be used in passive radar systems where the illuminating signal duration and bandwidth are content-dependent.
While the topic is interesting, there are some problems hindering the quality. I have some concerns with the current state of the manuscript.
1. In the keywords, the full name and abbreviation of PBR and PBL could be combined, like “Passive bistatic radar (PBR); passive coherent location(PCL)”.
2. The title of Figure 1 is too long. It is recommended to mark the meaning of the characters in the figure or explain them in the original text instead of explaining it in the figure title.
3. There are some figures in this paper that are unclear. For example, in Figures 5 to 7, the meaning of BWP is not explained in the paper, and the color distinction of the picture is poor. It seems that Figures 15 and 16 are not mentioned at all in the text. It is recommended to add some description of the figures or remove them directly to avoid misunderstandings. The legend of Figure 23 is missing.
4. The title of the section 5 is “Rényi entropy”, it is too general to describe the method proposed in this paper.
5. In section 5, the authors declare that “this article uses Rényi entropy to automatically select a signal that allows optimization of the radar equation”. However, there is very little description of this process in the article. This section mainly explained how to use the existing method to calculate the entropy of the 5G signal.
6. The proposed Rényi entropy based data processing method is inadequately described. I got confused while reading the paper. After calculating the entropy, did the method choose a right moment to detect the target? Or did the method make the signal meet the radar resolution requirement by integration for a long enough time? Can you describe it in more detail?
7. In real scenario, if the 5G signal does not always meet the requirements, does it mean that the target cannot be continuously observed? Is there a way to improve that?
8. The Section 6 is too brief and the structure is questionable. It is recommended to put the simulation parameter settings and some descriptions in this section instead of the section 5.
9. On the line 480 of page 13, it mentioned the drone is made of plastic. What’s the RCS of drone? Is it feasible in the actual experiment?
10. Last but not least, there are some language or formatted problems could be improved. For example, the table titles are generally above the table, but in this paper they are all below the table; The line number markers for Section 5 are gone (from page 10 to 12); In Section 5, the STFT is misspelled as “short-short Fourier transform”, it should be “short-time Fourier transform”; In section conclusion, “in this paper”, “in the paper” repeats. It is better for authors to recheck the full paper to avoid similar errors.
Reviewer 2 Report
Page 17, line 327, typographical error "ca".
Reviewer 3 Report
The authors should address the following issues:
1) The authors should clearly state the major contributions of their work in contrast to the existing state-of-the-art methods using 5G Networks in the literature such as the works in [15], [R1] and [R2].
[R1] P. Samczyński et al., "5G Network-Based Passive Radar," in IEEE Transactions on Geoscience and Remote Sensing, vol. 60, pp. 1-9, 2022, Art no. 5108209, doi: 10.1109/TGRS.2021.3137904.
[R2] Jingcheng Zhao, Xinru Fu, Zongkai Yang, Fengtong Xu, "Radar-Assisted UAV Detection and Identification Based on 5G in the Internet of Things", Wireless Communications and Mobile Computing, vol. 2019, Article ID 2850263, 12 pages, 2019. https://doi.org/10.1155/2019/2850263
2) Performance of the proposed method should be compared with the existing state-of-the-art methods using 5G Networks in the literature. For example, the results should be compared with the method in [15] , [R1] and [R2].
3) The proposed method is using the γ-order Rényi entropy defined in Equ. (8). The authors have used a fixed value of γ which is 3 in the simulations. However, the authors have not provided any procedure on how to select this value of γ. Also, the authors should investigate the impact of different values of γ on the performance of the proposed algorithm.
4) In the first line of the Abstract, the authors stated that "This paper presents the first successful drone detection results using a 5G network...". How this claim is justified as the work in [R2] already provided UAV detection using 5G networks?
5) The English writing of the manuscript can be improved.
Round 2
Reviewer 3 Report
I am satisfied with the authors' response.